

# Timing and ecological priority shaped the diversification of sedges in the Himalayas

Uzma[1,2,3], Pedro Jiménez-Mejías[4], Rabia Amir[1], Muhammad Qasim Hayat[1] and Andrew L. Hipp[2,3]

[1] Plant Systematics and Evolution Laboratory, Department of Plant Biotechnology, Atta-Ur-Rahman School of Applied Biosciences (ASAB), National University of Sciences and Technology (NUST), Islamabad, Pakistan
[2] Herbarium, The Morton Arboretum, Lisle, IL, United States of America
[3] Pritzker DNA laboratory, The Field Museum, Chicago, IL, United States of America
[4] Department of Biology (Botany), Universidad Autónoma de Madrid, Ciudad Universitaria de Cantoblanco, Calle Francisco Tomás y Valiente, Madrid, Spain

Corresponding authors
Uzma, uzma_165@yahoo.com
Pedro Jiménez-Mejías, pjimmej@gmail.com

## ABSTRACT

**Background**. Diversification patterns in the Himalayas have been important to our understanding of global biodiversity. Despite recent broad-scale studies, the most diverse angiosperm genus of the temperate zone—*Carex* L. (Cyperaceae), with ca. 2100 species worldwide—has not yet been studied in the Himalayas, which contains 189 *Carex* species. Here the timing and phylogenetic pattern of lineage and ecological diversification were inferred in this ecologically significant genus. We particularly investigated whether priority, adaptation to ecological conditions, or both explain the highly successful radiation of the Kobresia clade (ca. 60 species, of which around 40 are present in the Himalayas) of Himalayan *Carex.*

**Methods**. Phylogenetic relationships were inferred using maximum likelihood analysis of two nuclear ribosomal DNA (nrDNA) regions (ITS and ETS) and one plastid gene (*mat*K); the resulting tree was time-calibrated using penalized likelihood and a fossil calibration at the root of the tree. Biogeographical reconstruction for estimation of historical events and ancestral ranges was performed using the dispersal-extinction-cladogenesis (DEC) model, and reciprocal effects between biogeography and diversification were inferred using the geographic state speciation and extinction (GeoSSE) model. Climatic envelopes for all species for which mapped specimen data available were estimated using climatic data from WORLDCLIM, and climatic niche evolution was inferred using a combination of Ornstein-Uhlenbeck models of shifting adaptive optima and maximum likelihood inference of ancestral character states under a Brownian motion model.

**Results**. The Himalayan *Carex* flora represents three of the five major *Carex* clades, each represented by multiple origins within the Himalayas. The oldest *Carex* radiation in the region, dating to ca. 20 Ma, near the time of Himalayan orogeny, gave rise to the now abundant Kobresia clade via long-distance dispersal from the Nearctic. The Himalayan *Carex* flora comprises a heterogeneous sample of diversifications drawn from throughout the cosmopolitan, but mostly temperate, *Carex* radiation. Most radiations are relatively recent, but the widespread and diverse Himalayan *Kobresia* radiation arose at the early Miocene. The timing and predominance of *Kobresia* in high-elevation Himalayan meadows suggests that *Kobresia* may have excluded other

*Carex* lineages: the success of *Kobresia* in the Himalayas, in other words, appears to be a consequence largely of priority, competitive exclusion and historical contingency.

## INTRODUCTION

The habitat and topographic diversity of mountains make them important centers of biodiversity and rare species endemism (*Hughes & Atchison, 2015*; *Luo et al., 2016*; *Myers et al., 2000*; *Xie et al., 2014*; *Xing & Ree, 2017*; *Yu et al., 2015*; *Zhang et al., 2014*). Topographic diversity of montane regions is expected to provide opportunities for *in situ* diversification and genetic differentiation (*Holzinger et al., 2008*; *Liu et al., 2016*; *Matteodo et al., 2013*; *Pellissier et al., 2010*; *Villaverde et al., 2015a*; *Wang & Bradburd, 2014*). Numerous studies have highlighted the importance of mountains in the assembly and origin of some of the vital global biodiversity hotspots, such as the Great Cape region (*Richardson et al., 2001*), the Andes (*Hoorn et al., 2010*), and Madagascar (*Vences et al., 2009*).

At the same time, mountain ecosystems are isolated and limited in range, island-like (*Carlquist, 1965*; *Gehrke & Linder, 2009*; *Hughes & Eastwood, 2006*), making their biota particularly sensitive to climate change or other environmental perturbations. High-alpine lineages often have only long-distance dispersal and adaptation as options for responding to climate change. Recent studies have reported long-distance dispersals among mountains to be more frequent than formerly assumed (*Heaney, 2007*; *Levin, 2006*; *Schaefer, Heibl & Renner, 2009*; *Villaverde et al., 2015a*). But as global warming threatens to drive many cold-adapted species upslope toward extinction (*Chen et al., 2009*; *Thomas et al., 2004*; *Morueta-Holme et al., 2015*; though see *Crimmins et al., 2011* for an alternative to this traditional climate change scenario), research on the dynamics of species diversification in mountain systems is increasingly relevant.

A significant portion of the world's alpine diversity appears to have originated in the Pliocene and Pleistocene in the wake of late Miocene global cooling (*Herbert et al., 2016*; *Milne & Abbott, 2002*). However, some radiations triggered by the uplift of major mountain ranges seem to be older (*Hughes & Atchison, 2015*). For example, while the Tibetan biota exhibits abundant recent diversification (between 0.5 and 15 Ma; *Renner, 2016*), the effects of Tibetan Plateau uplift are reflected in much older divergences in the mid-Eocene. By contrast, the cold-adapted biota of the Tibetan Plateau didn't exist before climatic cooling from 13 Ma (Mid-Miocene) onward (*Favre et al., 2015*). Similarly, uplift of the neighboring Hengduan Mountains had a major effect on *in situ* diversification of resident lineages traceable to ca. 8 Ma (*Xing & Ree, 2017*).

The Himalayas, which adjoin both the Tibetan Plateau and the Hengduan Mountains, are the world's highest and one of its youngest mountain ranges, encompassing a wide ecoclimatic range (*Dobremez, 1976*). The rapid orogeny of the Himalayas, which began

ca. 59–50 Ma, continues today at a rate of ca. 5 cm yr $^{-1}$ (*Klootwijk et al., 1992*; *Patriat & Achache, 1984*; *Wang et al., 2012*), influencing the topography of adjoining mountain ranges (*Rolland, 2002*; *Xie et al., 2014*). The Himalayan range harbours ca. 10,500 plant species from 240 families (*Rana & Rawat, 2017*). This high variation in topography and vegetation types (*Mani, 1978*) makes this range one of the world's biodiversity hotspots. Moreover, the Himalayas constitute a vast cordillera, extending over 2,500 km from the border between Afghanistan and Pakistan in the west to northern Burma and western Yunnan in the east (*Searle, 2007*), which has had substantial effect on global climate through creation of monsoon conditions in Southeast Asia and the formation of xeric habitats in Central Asia (*Wan et al., 2007*; *Wang et al., 2012*; *Favre et al., 2015*). The Himalayas and adjacent regions are an excellent model for the study of ecologically driven continental species radiations (*Acharya et al., 2011*; *Grytnes & Vetaas, 2002*; *Korner, 2000*; *Price et al., 2011*).

*Carex* L. (Cyperaceae Juss.) is one of the largest angiosperm genera at ca. 2100 species (*Global Carex Group, 2015*; *Léveillé-Bourret, Starr & Ford, 2018*) and disproportionately important in the Himalayas, which harbours an estimated 189 species. Four major clades identified in the genus include the core Unispicate, Schoenoxiphium, Vignea and core Carex clades, which together are sisters to the smaller Siderosticta clade, comprising section *Siderostictae* and allied species (*Global Carex Group, 2016*; *Starr & Ford, 2008*; *Starr, Janzen & Ford, 2015*; *Waterway, Hoshino & Masaki, 2009*). *Carex* (including the previously segregated genera *Kobresia* and others; *Global Carex Group, 2016*) has a nearly cosmopolitan distribution, being present on all continents (*Hipp et al., 2016*), with a center of diversity in the northern temperate regions (*Starr, Naczi & Chouinard, 2009*). The diversity of *Carex* in the Himalayas is concentrated in the eastern regions, which harbor 153 species and have higher plant biodiversity in general (*Xie et al., 2014*), in contrast with the 112 species of the western half (*Govaerts et al., 2018+*). 40 *Carex* species are endemic to the Himalayas. The majority of *Carex* diversity of the Himalayas is represented by species of the core Unispicate clade, in particular species formerly segregated into genus *Kobresia* (ca. 40 species). The diversity of this clade raises a question as to whether the core Unispicate clade owes much of its diversity to *in situ* diversification in the Himalayas.

In this study, we address four questions regarding the influence of the Himalayan uplift on the diversification of *Carex*: (1) Are the timing of lineage and niche diversification in *Carex* coordinated with uplift of the Himalayas? (2) Is the diversification rate of *Carex* in the Himalayas higher than the background rate of diversification for the genus? (3) Is the high diversity of the former genus *Kobresia* (hereafter we will use *Kobresia* in this paper) in the Himalayas due to preadaptation to high elevation conditions, clade age (priority effect), or a mixture of both? (4) Are the Himalayas a source of global sedge biodiversity? We address these questions using phylogenetic comparative approaches that estimate lineage diversification rates and the history of trait and biogeographic evolution, on a phylogeny representing nearly half of all known Carex species. Ours is the first study on the diversification of sedges in the Himalayas, and thus it contributes significantly to our understanding of the origins of biodiversity in this important region of the globe.

## MATERIALS & METHODS

### Study area and taxa selection

For this study, 55 specimens representing 19 species were collected from Western Himalayas range (lies in Pakistan) between 2011 and 2017 (Table S1A). The collecting areas were selected as mentioned in Flora of Pakistan by Kukkonen (*Kukkonen, 2001*) for the genus. All specimens collected by the lead author (Uzma) were submitted to the herbaria of The Morton Arboretum (MOR), Lisle, Illinois, USA and Pakistan Museum of Natural History (PMNH), Islamabad, Pakistan. Additionally, 27 specimens of 21 species received from E (Royal Botanic Garden Edinburgh), MO (Missouri Botanical Garden) and MSB (Botanishche Staatssammiung München) herbaria were also included (Table S1B), comprising a total of 40 species from the Himalayas from which the new sequences were obtained for this study. In addition, we retrieved sequences for 944 species distributed worldwide from the Global *Carex* Group (2016) data matrix (available at Dryad: http://dx.doi.org/10.5061/dryad.k05qb). The final dataset included 56% (105 species) of Himalayan taxa out of the total 189 Himalayan species (Table S2). Four outgroup species (*Eriophorum vaginatum*, *Scirpus polystachyus*, *Trichophorum alpinum* and *Trichophorum caespitosum*) were included from the tribe *Scirpeae* (*Léveillé-Bourret et al., 2014*) for phylogenetic analyses. All data and scripts for the analyses are deposited in GitHub (https://github.com/uzma-researcher/Himalayan-Carex-Diversification; https://github.com/uzma-researcher/Himalayan-Carex-Climatic-niche-Evolution).

### Molecular methods

Total genomic DNA was isolated from silica-dried leaves of collected specimens and dried leaves of herbarium specimens using DNeasy Plant Mini Kit (QIAGEN, Valencia, California, USA, catalog # 69106) in the laboratory at The Morton Arboretum, Lisle, USA. Amplifications of two nrDNA regions, the internal transcribed spacer (ITS) and 5′ end of the external transcribed spacer (ETS), were performed using primer pairs ITS-IF and ITS-4R (*Urbatsch, Baldwin & Donoghue, 2000*; *White et al., 1990*) and ETS-1F and 18S-R (*Starr, Harris & Simpson, 2003*) respectively. The chloroplast (cpDNA) region *mat*K was amplified using two primer pairs: matK-2.1F and matK-5R (Kew Royal Botanic Garden, http://www.kew.org/barcoding/protocols.html) in the first step and matKF-61 and matKR-673 (nested primers; Global *Carex* Group, 2016) in the second step with slightly increased in annealing temperature (from 45. 0 °C to 51.0 °C). The PCR reaction mixture of 25μL contained: 2.5 μL 10X MgCl$_2$-free Taq buffer, 2.5 μL MgCl$_2$, 1.25 μL DMSO, 0.25 μL BSA, 0.25 μL of each primer at 20 mM, 0.25 μL Taq DNA polymerase (1.25 units), and 1 μL of genomic DNA as a template in amplifications of each region. These regions were selected based on suitability for wide-scale as well as fine-scale phylogenetics in the genus (*Starr, Harris & Simpson, 2003*; *Starr, Naczi & Chouinard, 2009*; *Global Carex Group, 2016*). Amplification cycles for ITS and ETS regions followed the conditions mentioned by *Hipp et al. (2006)* with minor adjustments in annealing temperature to get appropriate amplicons. However, two-step amplification of *mat*K region involved PCR conditions: 95.0° for 1:00; 30 cycles of: 95.0° for 0:45, 45.0° (51.0° for internal region) for 0:45, 72.0° for 1:30; 72.0° for 3:00. Amplified regions were cleaned and then sequenced following the

**Table 1  Total number of taxa, alignment length/total character or sites, number of informative characters, and models of evolution for each DNA region studied in phylogenetic analyses.**

| DNA regions studied | ETS | ITS | *mat*K |
|---|---|---|---|
| Total number of taxa | 915 | 892 | 772 |
| Alignment length/total characters or sites | 777 | 783 | 520 |
| Number of informative characters | 501 | 359 | 190 |
| Models of evolution AIC | GTR+I+G | GTR+I+G | GTR+I+G |

conditions as described by *Begley-Miller et al. (2014)* at Pritzker DNA laboratory, The Field Museum, Chicago, USA.

## Phylogenetic analyses and time-calibrated molecular phylogeny

The resulting new sequences for ETS, ITS and *mat*K regions were edited and assembled in Geneious version 9.1.6 (Biomatters, Auckland, New Zealand, available from http://www.geneious.com). Sequences were aligned for each region with Global *Carex* Group (2016) sequences using MUSCLE (*Edgar, 2004*) as implemented in Geneious. The matrices for each region were trimmed or N-filled to maintain equal characters in all sequences. The best-fit models of molecular evolution were estimated based on the Akaike information criterion implemented in jModelTest2 v.2.1 (*Darriba et al., 2012*; Table 1). The three DNA regions (ETS, ITS and *mat*K) were first analyzed separately using maximum likelihood (ML) method as implemented in RAxML HPC-PTHREADS-SSE3 version 8.2.4 (*Stamatakis, 2006*) with model of evolution GTR+I+G (General Time Reversible model with gamma distributions and invariant sites) and node support through 1000 rapid (fast) bootstraps to assess congruence among these nuclear and chloroplast DNA regions. The three matrices (ETS, ITS and *mat*K matrices) were then concatenated into a single matrix (with gaps or missing data). The combined matrix was analyzed using ML in RAxML with model of evolution GTR+I+G and 1,000 fast bootstraps to assess the phylogenetic relationship of the Himalayan *Carex*. These phylogenetic analyses were performed using the CIPRES Science Gateway v3.3 platform (*Miller, Pfeiffer & Schwartz, 2010*).

The Himalayan *Carex* species were non-monophyletic within each of three major clades (Vignea, core Unispicate, core Carex) in the resulting phylogenetic tree. Therefore, the Shimodaira-Hasegawa (SH) test (*Shimodaira & Hasegawa, 1999*) was performed as implemented in RAxML (version 8.2.4) to evaluate how strongly monophyly of Himalayan lineages was rejected, using five separate constraints that represent alternative scenarios for partial or complete monophyly of Himalayan lineages (Fig. S1; Table S3). Our null hypothesis ($H_0$) for each test was polyphyly of Himalayan lineages within the major clade (or taxon set) being tested, while $H_a$ was monophyly of Himalayan lineages within that clade. The ML tree generated under each constraint was compared to the unconstrained ML tree.

Divergence times were estimated on the ML tree (constructed on the concatenated dataset) after excluding outgroups using penalized likelihood (*Sanderson, 2002*) as implemented in treePL (*Smith & O'Meara, 2012*), which is designed for large phylogenies (*Spalink et al., 2016a*). We followed *Jiménez-Mejías et al. (2016)* and employed *Carex*

*colwellensis* Chandler which was the oldest reliable known fossil ascribable to *Carex*, dated back to the Priabonian, late Eocene. The crown node was fixed at 37.8–33.9 Ma according to previous reconstructions by *Míguez et al. (2017)*, an age compatible with *Carex colwellensis* (*Jiménez-Mejías & Martinetto, 2013*; *Jiménez-Mejías et al., 2016*). Penalized likelihood calibration was performed after optimizing priming and smoothing values using a $\chi^2$ test and cross validation. We assessed run convergence using the "thorough" option and inspected the tree visually using FigTree v1.4.2. The SH test and treePL analyses were performed using supercomputer facility at Research Center for Modeling and Simulation (RCMS), National University of Sciences and Technology (NUST), Islamabad, Pakistan.

## Biogeographical distribution and coding

For biogeographical analyses, we considered division of species into two groups: Himalayan and non-Himalayan, according to their presence or absence in the Himalayas (Table S2). Species distributions were based on the World Checklist of Selected Plant Families (*Govaerts et al., 2018+*), Flora of Pakistan (*Kukkonen, 2001*) and Flora of China (*Dai et al., 2010*). However, many species present predominantly in the Himalayas are also present in adjacent regions, making it difficult to study diversification patterns within the mountains. Therefore, we assessed sensitivity of our analyses to alternative biogeographic coding strategies by coding taxa in two different ways: "narrow sense" biogeographic coding, which treats the Himalayas as following strict Himalayan boundaries; and "broad sense" biogeographic coding, which includes the adjoining mountains (Karakoram, Hindu Kush, Tibet plateau, and Hengduan mountains) as part of a broadly construed Himalayas (Table S2). Secondly, to estimate ancestral ranges at the global scale, we scored taxa according to ten ecozones/biogeographical realms based on published distributions. However, our coding diverges from traditional coding (*Udvardy, 1975*) in two regards: (1) we split the Palearctic region into Western and Eastern to separate Europe, north Africa and western Asia from eastern and central Asia; and (2) we treat the Himalayas in the broad sense as a tenth region (biogeographic coding in Table S4).

Georeferenced data for each *Carex* species available at GBIF data portal (http://www.gbif.org/) were retrieved using species synonyms to gather all records under their current taxonomic names (*Global Carex Group, 2015*), listed in Table S5. We downloaded GBIF data using the rgbif package (*Chamberlain, 2017*) in R (*R Core Team, 2013*). The largest number of records (17,841) was retrieved for species *Carex lasiocarpa*, followed by *Carex canescens* with 16,825. 41 species had only one record and 324 species had fewer than 50 records. We cleaned data to remove specimens georeferenced outside the reported range of the species (according to *Govaerts et al., 2018+*) and eliminated duplicates using the dismo (*Hijmans et al., 2011*), ape (*Paradis & Schliep, 2018*) and magrittr (*Bache & Wickham, 2016*) packages in R. Species data were also thinned to exclude records within 1.0 to 1.6 km of each other (following *Hipp et al., 2018*). The post-thinning dataset comprised 965,556 occurrence records for 850 unique species, each represented by an average of 1,136 specimens.

## Geographic-dependent diversification and extinction

The GeoSSE model (*Goldberg, Lancaster & Ree, 2011*) as implemented in the diversitree R package (*FitzJohn, 2010*) was used to assess historical differences in the rate of speciation, dispersal and extinction in the Himalayas versus non-Himalayan regions in narrow sense (total 966 species of which 10 are endemic to the Himalayas, 861 are endemic to non-Himalayas, 95 are present in both regions) and broad sense (51 endemic to Himalayas, 838 to non-Himalayas, 77 species present in both regions) biogeographic coding (using Table S2 for geographic states coding to narrow and broad sense coding and resulting dated phylogeny). Diversification rate was estimated in three states: A representing species endemic to the non-Himalayas, B representing species endemic to the Himalayas, and AB representing species present in both regions. In this model, we estimate speciation rates of species in region A (sA) and B (sB), as well as sAB, speciation rates for taxa that give rise to two daughter species, one in each region. Likewise, geographical range expansion from A or B to AB were estimated with rates of dispersal dA and dB respectively and range contraction with rates of extinction xA and xB.

Full and constrained alternative GeoSSE models were tested: the full model, in which rates of speciation, extinction and dispersal differed among two regions; a constrained model in which speciation in taxa distributed across two regions was set to zero (sAB $\sim$ 0); and a model in which rates of speciation and extinction were constrained to be the same between regions (sA $\sim$ sB; xA $\sim$ xB). Models were compared using the Akaike Information Criterion. The best-fit model was also fitted using Bayesian Markov chain Monte Carlo (MCMC) for 100,000 generations and an exponential prior to estimate parameter distributions.

## Biogeographical reconstruction and ancestral range estimation

Biogeographic history was investigated as transitions among the ten regions initially coded, excluding Antarctica and Oceania for computational reasons. The remaining regions were the Himalayas, Indo-Malaya, Eastern Palearctic, Western Palearctic, Nearctic, Afrotropic, Neotropic and Australasia. The DEC model was utilized as implemented in Lagrange (*Ree et al., 2005*; *Ree & Smith, 2008*) and BioGeoBEARS (*Matzke, 2014*) using BSM (Biogeography stochastic mapping). The DEC method implements the maximum-likelihood approach and allows vicariance, range expansion (dispersal) and range contraction (extinction) processes with inclusion of different parameters. The BSM analysis is based on the Bayesian MCMC approach and simulates the biogeographical history of the events (anagenetic and cladogenetic events) along the branches of the tree. In the BSM analysis, in total 1,000 stochastic mapping replicates with 50,000 maximum trees per branch were conducted on ML tree. The alternative biogeographic model implemented in BioGeoBEARS was not considered, as the inclusion of the jump parameter is not directly testable relative to non-jump models (*Ree & Sanmartín, 2018*) and introduces complexities that are not necessary to explain our biogeographic scenarios.

## Climatic niche modelling

To characterize climatic envelopes for the species for which specimen data were available, we extracted 19 bioclimatic variables from WORLDCLIM data using the R package raster

(WorldClim v1.4; *Hijmans et al., 2005*). The average values for species occurrence data and 19 bioclimatic variables from a total of 965,556 data records for 850 unique species were estimated. The obtained data for 838 species were further proceeded after removing outliers from WORLDCLIM data. For each bioclimatic variable, ancestral character states were estimated assuming a Brownian motion trait evolution process using the 'fastAnc' function in phytools (*Revell, 2012*). Then non-metric multidimensional scaling (NMDS) ordination was employed using Euclidean distances normalized to unit variance on bioclimatic data for the tips as well as the internal nodes of the tree. The stress from $K = 1$ to $K = 10$ was calculated and plotted against dimension to estimate how much additional information was extracted from the bioclimatic data with each additional NMDS axis.

Data were plotted with internal nodes of interest, representing ancestors of Himalayan radiations, colored for identification. The Himalayan taxa of the core Unispicate clade are represented by two major subclades, with *Kobresia* (40 species) sister to a clade dominated by *Uncinia* (30 species). Placement of the *Kobresia* ancestors were compared with ancestors of all other Himalayan radiations to evaluate whether the *Kobresia* ancestors may have represented an ecologically specialized (preadapted) species to Himalayan climatic conditions. Alternatively, the earlier origin of *Kobresia* species in the Himalayas would support clade age (priority effect and possibly competitive exclusion) as an explanation for success of *Kobresia* relative to other more recent Himalayan radiations. Niche analyses were conducted using the phytools and vegan (*Oksanen et al., 2017*; *Revell, 2012*) packages in R.

However, the analysis above presupposes a Brownian motion model of niche evolution, which is unrealistic in the face of natural selection (*Butler & King, 2004*; *Hansen, 1997*). Underparameterizing the model of niche evolution risks inflating Type II error (incorrectly failing to reject the null hypothesis of no difference among clades) if clades tend to adapt to new adaptive regimes to which species migrate. To address this limitation, we also analyzed our niche data using a multivariate Ornstein–Uhlenbeck (O-U) model implemented in the PhylogeneticEM package of R to test transitions in selective regime in an information theoretic framework (*Bastide, Mariadassou & Robin, 2017*; *Bastide et al., 2018*). Analyses were conducted using a scalar O-U process and default search settings for the PhyloEM function.

## RESULTS

### Phylogenetic relationship of Himalayan species and estimation of Himalayan clade divergence in tribe *Cariceae*

New sequences obtained for this study are deposited in GenBank (Table S6), and details on each of the DNA matrices are reported in Table 1. The *mat*K tree topology (File S1 for matrix) showed low support for most relationships (File S2). Similarly, the phylogeny obtained from ITS (File S3 for ITS matrix) showed phylogenetic incongruence with previous studies (*Global Carex Group, 2015*; *Waterway, Hoshino & Masaki, 2009*; *Waterway et al., 2016*), with low support for the major clades (File S4). The ETS region (File S5 for matrix) was mostly well-defined for major clades with high to moderate supports (File S6). The

**Table 2** Divergence time estimates based on penalized likelihood calibration.

| Clades | Number of species in each clade | Crown age (Ma) ca | Himalayan clades divergence (Ma) ca | Geological time period |
|---|---|---|---|---|
| Siderostictae | 6 | 17.2 | | |
| Schoenoxiphium | 24 | 19.4 | | |
| Vignea | 236 | 23.8 | 7.2 | Late Miocene |
| Core Unispicate | 132 | 24.0 | 20.6 | Early Miocene |
| Core Carex | 571 | 22.9 | 14.4 | Early Miocene |
| Core Unispicate -core Carex | 703 | 26.5 | | |

three matrices with least missing data for one individual per species were concatenated into a single matrix of 970 sequences (outgroups included) with 2,080 sites (File S7), containing 48.7% missing data coded as gaps. The topology of the tree based on combined DNA datasets was highly congruent with previous studies (*Global Carex Group, 2015*; *Waterway, Hoshino & Masaki, 2009*; *Waterway et al., 2016*) and supported at most major clades. All the four major clades (Schoenoxiphium, Vignea, core Unispicate and core Carex) sister to Siderostictae clade were recovered (File S8). The Himalayan species (105) fall into three major clades of *Carex*: (1) Vignea, (2) core Unispicate and (3) core Carex. The Himalayan taxa (16 species) in the Vignea clade were largely spread out in the clade, not sister to other Himalayan taxa, though two sister species pairs were observed. Himalayan species in core Unispicate clade (30 species) mainly belong to *Kobresia*, except for two unrelated species formerly classified under section *Leucoglochin*, arrayed in two clades: Kobresia clade 1, which exhibited only three species (minor clade), and Kobresia clade 2 (major clade) with 27 Himalayan species (File S8). Himalayan taxa (59 species) in core Carex occurred as small clades or isolated (single) species dispersed throughout the clade. Our topology may be poorly supported in some clades, but the groups and branching we retrieved was fully compatible with the *Global Carex Group (2016)* tree, where the clades retrieved were more strongly supported.

Date calibrations (Fig. S2) are also congruent with previous reports (*Escudero et al., 2012*; *Waterway, Hoshino & Masaki, 2009*; *Waterway et al., 2016*), which is expected given that they are based on the same sources (performed on ML tree). The earliest Himalayan diversification is the *Kobresia* radiation in the core Unispicate clade at the early Miocene (around 20.6 Ma), followed by diversifications in core Carex (early Miocene, 14.4 Ma) and Vignea (late Miocene, 7.2 Ma; Table 2) clades. Although all Himalayan taxa diversified in three major clades in the epoch Miocene, the timing of diversification of Himalayan *Carex* was different in three major clades, which suggested multiple origins and also non-monophyletic lineages. Our time calibrations do differ from recent work (*Spalink et al., 2016a*; *Spalink et al., 2016b*) due to the fact that we use a calibration based on a more recent date, compatible with the fossil record (*Jiménez-Mejías et al., 2016*).

Monophyly of the Himalayan taxa within each major clade was strongly rejected at the 0.01 level based on the Shimodaira-Hasegawa test, except for the *Kobresia* species in the core Unispicate clade, where monophyly cannot be rejected (Table 3). We focused an
**Table 3  The Shimodaira-Hasegawa (SH) tests, evaluating monophyly of Himalayan lineages.**

| Topologies of False tree | lnL, constrained (A) | lnL, unconstrained (B) | delta lnL ($\delta$) = (A)–(B) | s.d. | P |
|---|---|---|---|---|---|
| Constraint tree 1 | −24,701.56 | −23,096.69 | −1,605.48 | 104.30 | $P < 0.01$ |
| Constraint tree 2 | −17,317.09 | −16,888.41 | −427.60 | 46.52 | $P < 0.01$ |
| Constraint tree 3 | −52,849.70 | −50,206.00 | −2,644.64 | 167.53 | $P < 0.01$ |
| Constraint tree 4 | −84,061.72 | −79,483.50 | −4,577.86 | 231.26 | $P < 0.01$ |
| Constraint tree 5 | −7,496.76 | −7,495.88 | −1.32 | 10.96 | non-significant |

Notes.
Constraint tree 1 = Himalayan species in Vignea clade are monophyletic.
Constraint tree 2 = Himalayan species in core Unispicate clade are monophyletic.
Constraint tree 3 = Himalayan species in core Carex clade are monophyletic.
Constraint tree 4 = Himalayan species in core Carex, core Unispicate and Vignea clades are monophyletic.
Constraint tree 5 = All *Kobresia* species are monophyletic.

additional test on the monophyly of *Kobresia*, in which *Kobresia* was constrained to be monophyletic and only the Unispicate taxa included. In this test, monophyly of *Kobresia* was not significantly rejected ($\Delta$lnL = −1.32). Thus, while *Kobresia* was found to be polyphyletic in previous work (*Global Carex Group, 2016*) and in the current study, we considered this result poorly supported by the data, pending additional study.

## Ancestral range reconstruction

Initial biogeographic reconstruction under the DEC model was conducted on the "narrow sense" (File S9) biogeographic coding, which failed to recover any clade endemic to the Himalayas, which is at odds with our observation that numerous species are predominantly Himalayan in origin. We thus present here analysis of data coded in the "broad sense" (see methods; File S10), using the stochastic mapping (BSM) method on the DEC model as implemented in BioGeoBEARS. We generated 1000 stochastic maps in every 50,000 trees per branch in BSM analysis. In each stochastic map, total event counts of cladogenetic events were higher which were based on vicariance (5.6% events) and sympatric (94.4% events) processes than that of anagenetic which were based on only dispersal events (Table S7). However, an interesting finding in BSM results was that none of the events showed "range switching dispersal", while all the observed dispersal was of "range expansion".

Under the broad sense biogeographic coding through BSM, 128 Himalayan taxa were distributed among three major clades: (1) Vignea (16%), (2) core Unispicate (29%) and (3) core Carex (54.6%) (Fig. 1). The highest number of dispersal events into the Himalayas was obtained from Eastern Palearctic (Mean = 39.42; s.d. = 5.23) followed by Nearctic (Mean = 21.44; s.d. = 4.31) and Western Palearctic (Mean = 14.48; s.d. = 3.63) regions. The lowest number of dispersal events was obtained from three regions into the Himalayas; Neotropic (Mean = 3.60; s.d. = 2.47), Australasia (Mean = 3.64; s.d. = 1.83) and Indo-Malaya (Mean = 5.22; s.d. = 2.16). The *in situ* diversification within the Himalayas that precedes dispersal from the Himalayas region correspondingly into the Eastern Palearctic with the maximum average dispersal events (Mean = 35.26; s.d. = 4.11), however, the following two regions were Indo-Malaya and Western Palearctic (Means = 12. 88 and 12.48, s.d. = 3.31 and 3.72, respectively) (Table S7).

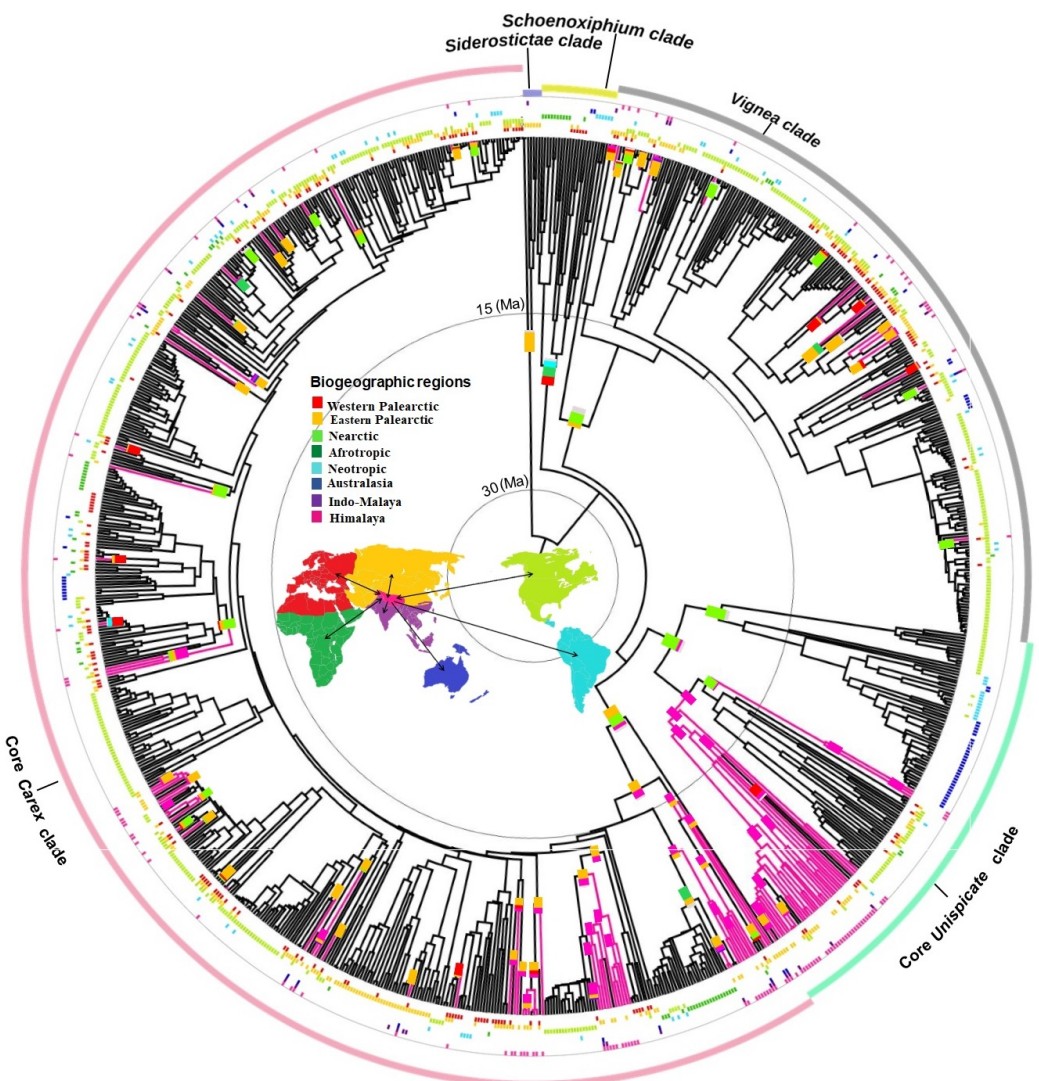

**Figure 1** **Biogeographical reconstruction of Himalayan *Carex*.** DEC analysis on broad sense dataset used dated phylogeny (966 species and times in millions of years (Ma)) representing biogeographical history of ancestral ranges in Himalayan *Carex* lineages designated in three major clades (Vignea, core Unispicate and core Carex). Here at tips of the branches, geographical ranges for extant taxa are labeled, while the outer colored circles represent five major clades (Siderostictae, Schoenoxiphium, Vignea, core Unispicate and core Carex). The colored blocks at internal nodes show ancestral ranges. The pink colored branches designate Himalayan taxa. The map denotes 8 biogeographic ranges with color scheme (Western Palearctic, Eastern Palearctic, Nearctic, Afrotropic, Neotropic, Australasia, Indo-Malaya, Himalaya).

The oldest colonization into the Himalayas, at the crown of the Kobresia clade (comprising 37 Himalayan species out of 128 species in total based on broad sense), occurred during the early Miocene (ca. 20 Ma). Taxa from the Vignea clade originate from the Nearctic and Western Palearctic and Eastern Palearctic (Fig. 1). All origins are reconstructed as arising from relatively recent colonization of the Himalayas (started in late Miocene, ca. 7.2 Ma and last colonization appeared very recently at ca. 0.09 Ma and ca. 0.35

Ma by *Carex physodes* and *Carex canescens*, respectively). In the core Unispicate and core Carex clades, by contrast, the Himalayan species cluster into around 12 small clades of 2 to 33 species (File S10). The Himalayan species from the core Unispicate clade derived mostly from Nearctic ancestors with one radiation deriving from the Eastern Palearctic (Fig. 1, File S10). The colonization out of the Himalayas were observed into the Eastern and Western Palearctic, Indo-Malaya, Neotropic, and Nearctic regions around 10 million years ago. While in core Carex, the Himalayan species are arranged in small sub-clades which arose around the early Miocene, there was more recent (ca. 8.5 Ma) *in-situ* diversification within the Himalayas that served as a source for other regions (File S10). The ancestral ranges of the Himalayan taxa were predominantly in the Eastern Palearctic, Western Palearctic and Nearctic regions. However, in core Carex, Himalayan taxa arise from these three regions as well as the Afrotropic region (Fig. 1). Further, the core Carex dispersals were inferred from the Himalayas into these four regions and particularly Australasia. However, our sampling was somehow biased towards the Nearctic region, this was a direct consequence of using the *Global Carex Group (2016)* dataset, which is the largest and most complete *Carex* dataset to date (>50% of the total diversity).

## Diversification dynamics of Himalayan species

Among the three GeoSSE models evaluated with AIC values, the best-fit model (Table 4) for the narrow ($\Delta$AIC = 49.94) and broad ($\Delta$AIC = 76.08) sense biogeographic coding was the constrained model in which speciation in taxa distributed across two regions was set to zero (sAB $\sim$ 0). Both biogeographic coding (narrow and broad) strongly reject the model constrained to have speciation rates (sA-sB) equal in the Himalayas and non-Himalayas and extinction rates also equivalent (xA-xB). Given our finding that the broad-sense biogeographic reconstruction recovers Himalayan lineages (see biogeography results above), we considered the broad-sense biogeographic coding to be better suited to estimate diversification rates within versus outside of the Himalayas (Table 4). Although the Himalayas showed positive net diversification rate (0.16 events per million years) there was no real difference in diversification rates between the two areas (0.06 events per million years). We utilized Bayesian MCMC on the broad sense dataset to estimate parameter uncertainty (Table 4, Fig. S3). Figure 2 represents posterior probability differences between the lineages of the two regions and diversification rates. It suggests the rate of speciation is indeed lower in the Himalayas than non-Himalayas with difference (0.34 events per million years). A similar finding was obtained with the rate of extinction (0.28 events per million years). The rate of dispersal was higher in Himalayan than non-Himalayan lineages with difference events per million years (0.27 events per million years); while not significant, this result suggests that independent origins of Himalayan taxa from non-Himalayan ancestors may be less common than colonization of non-Himalayas from Himalayan radiations (Fig. S3).

## Climatic niche evolution

Plotting stress against number of dimensions in a set of initial NMDS ordinations that include data for both the tip states and ancestral reconstructions (Fig. S4) shows significant

Uzma et al. (2019), *PeerJ*, DOI 10.7717/peerj.6792

**Table 4  Estimates of diversification in Himalayan vs non-Himalayan lineages using Geographic State Speciation and Extinction (GeoSSE) models.** In these models, the Himalayas construed broadly or narrowly (see methods) is denoted as area B; areas outside the Himalayas are denoted as area A.

| Biogeo-graphic coding | Models | -lnL | AIC | sA | sB | sAB | dA | dB | xA | xB | Net diversification rate | |
|---|---|---|---|---|---|---|---|---|---|---|---|---|
| | | | | | | | | | | | Region A sA-xA | Region B sB-xB |
| Narrow sense | Reduced full (sA, sB, dA, dB, xA, xB) | 2,673.88 | 5,361.76 | 0.50 | 0.16 | 0 | 0.01 | 0.60 | 0.30 | 1.00E–05 | 0.20 | 0.16 |
| | **Constrained 1** (sAB ∼ 0) | **2,673.88** | **5,359.76** | **0.50** | **0.16** | **0** | **0.01** | **0.60** | **0.30** | **0** | **0.20** | **0.16** |
| | Constrained 2 (sA ∼ sB, xA ∼ xB) | 2,699.9 | 5,409.79 | 0.49 | 0.49 | 9.00E–06 | 0.02 | 3.15 | 0.42 | 0.42 | 0.07 | 0.07 |
| Broad sense | Reduced full (sA, sB, dA, dB, xA, xB) | 2,715.04 | 5,444.09 | 0.50 | 0.16 | 9.00E–06 | 0.009 | 0.28 | 0.28 | 2.00E-06 | 0.22 | 0.16 |
| | **Constrained 1** (sAB ∼ 0) | **2,715.04** | **5,442.08** | **0.50** | **0.16** | **0** | **0.009** | **0.28** | **0.28** | **0** | **0.22** | **0.16** |
| | Constrained 2 (sA ∼ sB, xA ∼ xB) | 2,754.08 | 5,518.16 | 0.40 | 0.40 | 0 | 0.01 | 0.19 | 0.17 | 0.17 | 0.23 | 0.23 |

**Notes.**

lnL, log-likelihood; AIC, Akaike Information Criterion; sA, sB, speciation rate in area A, B; sAB, speciation rate in taxa whose distribution includes both areas A and B; dA, dB, dispersal from area A to B or B to A respectively; xA, xB, extinction rate in area A, B.
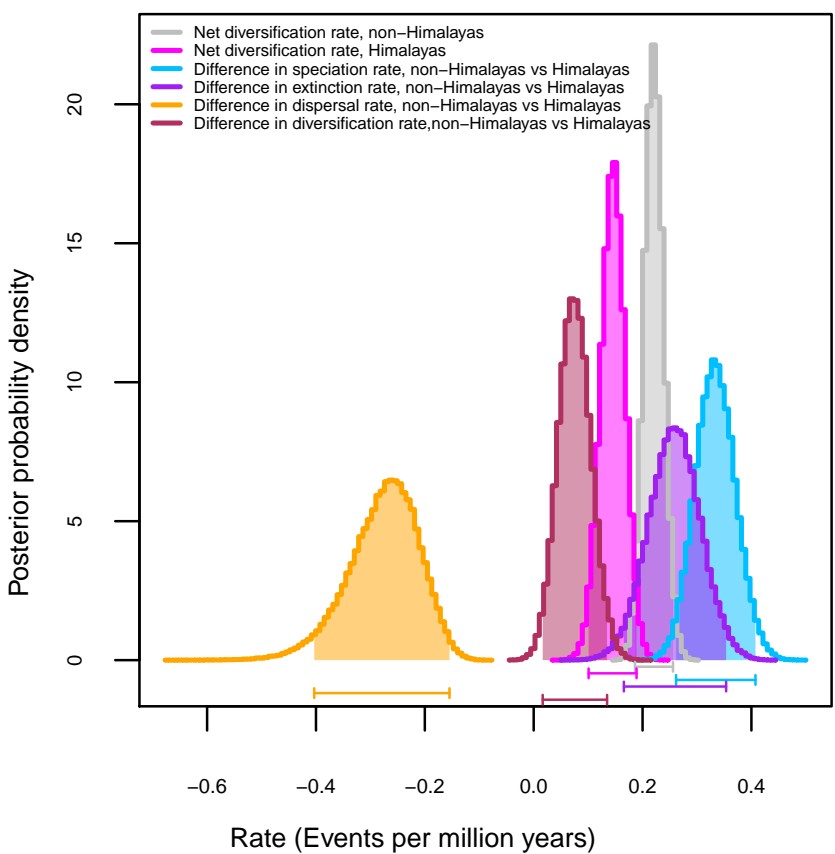

**Figure 2 Diversification rates for Himalayan *Carex* lineages estimated under the model Geographic State Speciation and Extinction (GeoSSE).** The differences in rates of speciation, extinction, and dispersal in Himalayan lineages verse non-Himalayan lineages, are estimated. The Himalayan lineages show higher dispersal rate than non-Himalayan species, however, speciation and extinction rates instead lower in Himalayan lineages compared to non-Himalayan species. The horizontal bars below each curve represent 95% confidence interval (CI) under MCMC.

decreases in stress up to $K = 5$. However, as visual inspection of the 5-dimensional ordination showed no qualitative differences from the 2-dimensional ordinations, ordination results here were presented for the $K = 2$ NMDS analysis. For ordination of both the inferred ancestral states (Fig. 3A) and the tip states (Fig. 3B), eight BIOCLIM variables correlate strongly ($|r| > 0.7$) with MDS axis 1 (BIO1, BIO4, BIO6, BIO9, and BIO11, which relate to temperature; and BIO12, BIO13, and BIO16, which relate to precipitation) and three with MDS axis 2 (BIO14 and BIO17, precipitation of the driest month and quarter respectively; and BIO15, precipitation seasonality, estimated as the coefficient of variation). In the ordination of inferred ancestral states, the two Kobresia clade ancestors were not significantly differentiated from the ancestors of the remaining Himalayan taxa. While the maximum likelihood estimator underlying these reconstructions was biased against detecting differentiation among tips closer to the base of the tree (because the estimator under a Brownian motion process is the weighted mean of all tips in the tree; (*Felsenstein, 1985*), the ordination of tip states (Fig. 3B) was not biased in this way, and

should show differentiation if the Kobresia clades are descended from an ancestor that was uniquely adapted to a different climatic niche. To the contrary, the 95% C.I. for the Kobresia clade climatic niche was almost entirely contained within the 95% C.I. for the remaining Himalayan taxa and exhibited a narrower variance, which was expected given that it reflected a narrower phylogenetic diversity (Fig. 3B). Both groups were also not significantly differentiated from the remainder of *Carex*.

Analysis using the multivariate scalar O-U model, on the other hand, recovered two major shifts in selective regime: one at the base of the Uncinia clade, which is strongly represented in the Neotropics and New Zealand; and one at the base of a clade comprising primarily of the traditional *Indicae* and *Decorae* sections, groups of primaril tropical distribution (Fig. 4) for the given data set employed in this study. These clades are characterized respectively by higher temperatures, higher precipitation, and lower seasonality (*Uncinia*); and higher temperatures, lower precipitation, lower temperature seasonality, and higher precipitation seasonality (*Indicae/Decorae* clade). Himalayan taxa overall are widespread in climatic niche and exhibit no consistent trend (Fig. S5).

## DISCUSSION

Our study demonstrates that Himalayan diversity of the cold-adapted *Carex* reflects a complex history of migrations and *in situ* diversification episodes rather than a few distinct radiations within the Himalayas. The dominance of the portion of the core Unispicate clade that includes *Kobresia*—the oldest radiation that we detected (early Miocene, ∼20 Ma)—is attributable to priority effect and represents historical contingency upon entry to the Himalayas. This major clade, combined with diversification of several minor clades and numerous dispersals into the Himalayas, explains the diversity of Himalayan sedges we observe today.

### Himalayan orogeny and the origin of the Kobresia clade

*Carex* taxa of the Himalayas exhibited high phylogenetic diversity both among clades—three of the five major *Carex* clades were represented in the Himalayas—and within clades, where the Himalayan *Carex* are mostly highly polyphyletic. The divergence time for the largest Himalayan *Carex* clade, the Kobresia clade, falls in the early Miocene (ca. 20.6 Ma). This postdates the initial uplift of the Himalayas by about 30–39 million years (see introduction above), during which time the Indian Tibetan continent collided with Asian plates. It corresponds well, however, with the second stage of Himalayan uplift (25–20 Ma), at which time accretion of the upper layer of the Indian continental crust further raised the Himalayas (*Molnar & Stock, 2009*, *Van Hinsbergen et al., 2011*; *Van Hinsbergen et al., 2012*). This combination of uplift events introduced the summer monsoon precipitation regime and alpine climates at altitudes of 5,000–6,500 m (*Xie et al., 2014*), which might have contributed to migration of floristic elements into the region. Our interpretation is that the crown diversification of the core Unispicate clade that includes the Kobresia clade was roughly simultaneous with the second Himalayan uplift event, and that the dominance of the Kobresia clade among Himalayan *Carex* is likely due in part to historical contingency:

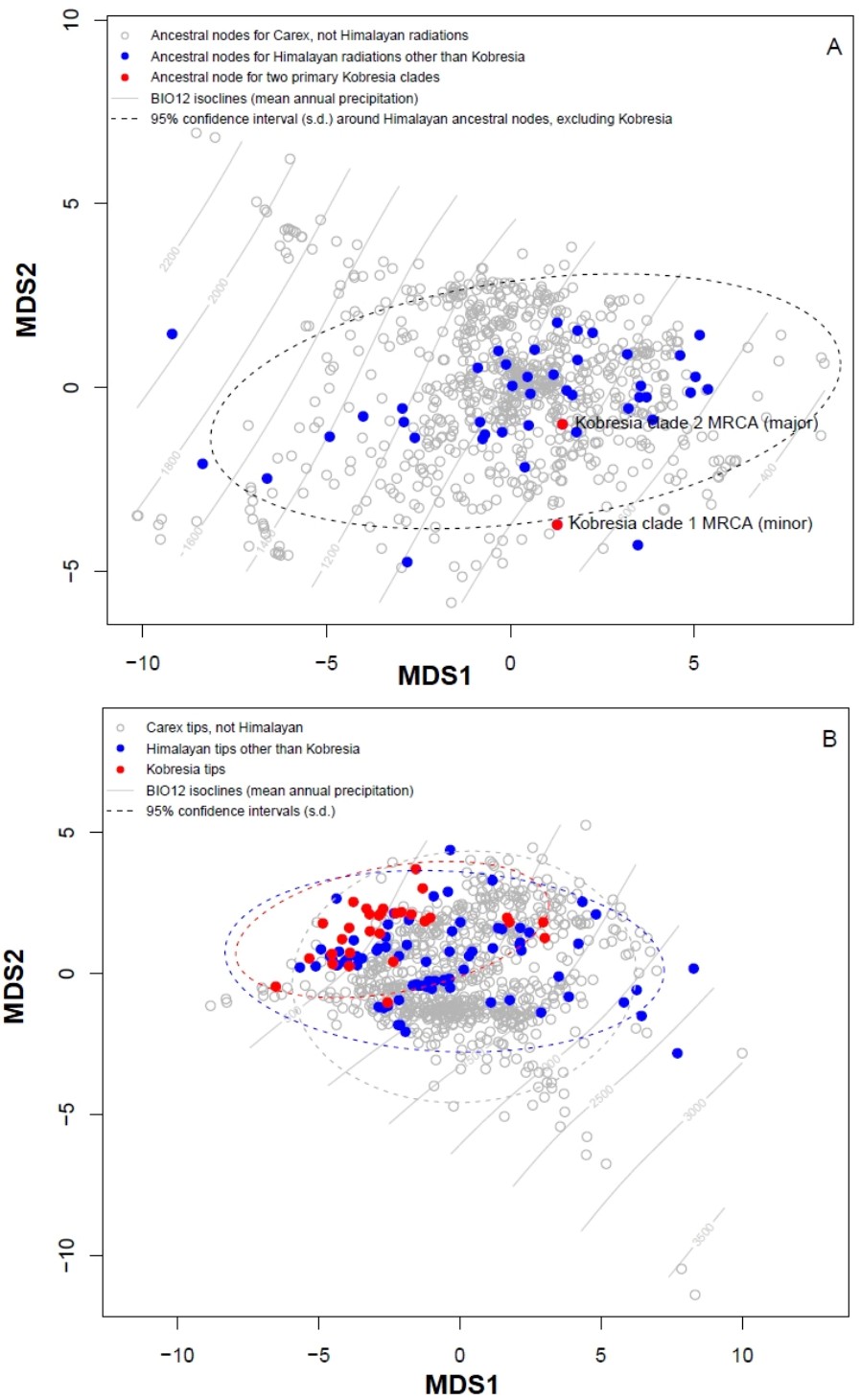

**Figure 3** **Multidimensional scaling (MDS) analysis to infer Himalayan *Carex* ancestral and tip states.** Ordination of ancestral states (A) and the tip states (B) of the Himalayan and non-Himalayan for bioclimatic variable BIO12, mean annual precipitation. Here the 95% confidence interval (C.I.) for the Kobresia clade climatic niche exhibits a narrower variance for the remaining taxa. 🖼 DOI: 10.7717/peerj.6792/fig-3

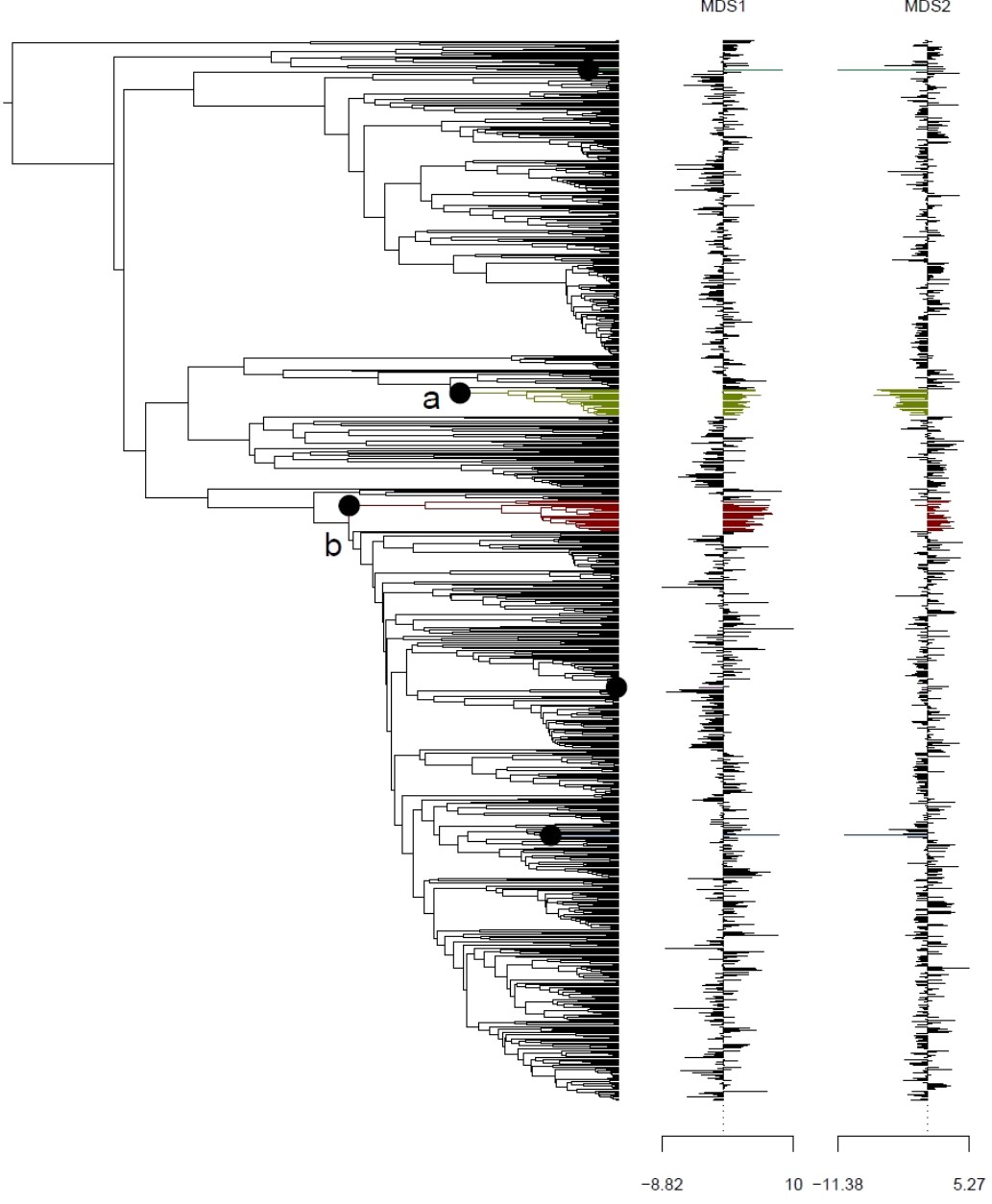

**Figure 4   Climatic niche evolution for range shift in Himalayan taxa using Ornstein-Uhlenbeck (O-U) model.** Species range shift in ecological space estimated on $K = 2$ MDS ordination using multivariate scalar O-U model. Here colors changes indicate significant transitions in climatic space and identifies two major shifts: (A) at the base of the Uncinia clade and (B) at the base of a clade comprised primarily of the traditional sections, *Indicae* and *Decorae*.

*Kobresia* simply arrived first to the high alpine Himalaya, excluding potential competitors rather than exhibiting particularly strong preadaptations.

The *Kobresia* radiation is singular in its diversity: the remaining Himalayan *Carex* species arose from 43 migrations into the Himalayas from two other major clades, in which all

Himalayan species in Vignea clade lacked Himalayan sisters except two clades of 2 species each; while core Carex Himalayan species also largely lacked Himalayan sisters (25 species), except 10 clades of 2–11 species. Based on age alone, we would expect *Kobresia* to exhibit higher species diversity than the other Himalayan radiations (*Wiens et al., 2009*). However, our analyses of multiple-regime Ornstein–Uhlenbeck models suggested that there are not special attributes of the Kobresia clade that make it successful in the Himalayas, but rather that its early arrival to the unique Himalayan climates enabled its success. Further, *Kobresia* exhibited no innate attributes that allowed it to take over the Himalayas. Therefore, our analyses suggested that possibly the contingency (resulting possibly due to historical abiotic events (*Fukami, 2015*), e.g., uplift of the region during orogeny of the Himalayas) in *Kobresia* assembly with the combination of clade age (priority effect) and competitive exclusion (*Abrams, 1983*; *Brown & Wilson, 1956*; *Grant & Grant, 2006*; *Monroe & Bokma, 2017*; *Schluter, 2000*) likely explain the success of the Kobresia clade and the floristic dominance of its species in the Himalayas when compared to other Himalayan *Carex* groups.

## Long-distance migration and multiple ancestral ranges

Previous biogeographic studies in *Carex* (e.g., *Escudero et al., 2009*; *Hoffmann, Gebauer & Rozycki, 2017*; *Miguez et al., 2017*; *Spalink et al., 2016b*; *Villaverde et al., 2015b*) have focused on specific clades or broader taxonomic scales. Our study utilizes the broadest phylogenetic sampling of the genus *Carex* to date (966 species, ca. 50% of the extant diversity; Fig. 1) to address the origins of a highly polyphyletic regional flora, the *Carex* of the Himalayas. The earliest origin of the Himalayan *Carex* flora was the Kobresia clade, which arose in the early Miocene from a Nearctic ancestor (Fig. 1), presumably by bird-mediated long-distance dispersal (cf. *Villaverde et al., 2017b*). Such long-distance migrations are well-documented in the genus under diverse scenarios (*Escudero et al., 2009*; *Jiménez-Mejías, Martín-Bravo & Luceño, 2012*; *Miguez et al., 2017*; *Villaverde et al., 2015a*; *Villaverde et al., 2015b*; *Villaverde et al., 2017a*; *Villaverde et al., 2017b*). There were few radiations out of the Himalayas in this clade to the adjacent Eastern Palearctic and Indo-Malaya as well as to more distant regions (Western Palearctic, Neotropic, and Nearctic). We detected no radiation into the Himalayas from the most closely adjacent region, Indo-Malaya (Fig. 1). Very likely this is due to the fact that Indo-Malaya is dominated by tropical and subtropical dry and moist broadleaf forests, and *Carex* is a predominantely temperate group (*Escudero et al., 2012*; *Spalink et al., 2018*). All these findings were based on the subset of taxa utilized in this study; inclusion of additional taxa might of course influence these results, but we have no reason to suspect that they are biased.

In the core Carex clade, by far the largest *Carex* clade globally (*Global Carex Group, 2015*; *Global Carex Group, 2016*), Himalayan species cluster into small clades (2–11 species) and 25 individual species that arose from outside the Himalayas. Ancestral origins for these clades and species are remarkably disparate (File S10). Thirty Himalayan species derive from the traditional sections *Graciles, Decorae, Indicae, Setigerae, Aulocystis, Thuringiaca, Polystachyae, Clandestinae* and *Radicales* derived primarily from Eastern Palearctic ancestors. Twelve Himalayan species from sections *Racemosae, Aulocystis, Vesicariae* and

*Phacocystis* derived from the Nearctic, while a smaller number of Himalayan species (4) from the traditional *Thuringiacae, Spirostachyae, Ceratocystis,* and *Hallerianae* derived from the Western Palearctic. The Afrotropics seemingly contributed only *C. obscuriceps* to the Himalayas (Fig. S3), although this inference may be due to undersampling of the large traditional section *Vesicariae*. It is similarly striking how many radiations are inferred to have given rise to geographically distant lineages. Dispersals from the Himalayas to the Afrotropic and Australasian region in core Unispicate and core Carex clades bear further investigation.

## CONCLUSION

Our analyses demonstrate that Himalayan diversification in the core Unispicate clade has contributed significantly to global sedge diversity. But they also suggest that diversification rates may have been similar within the Himalayas versus non-Himalayas (difference in diversification = 0.06). In contrast to studies demonstrating an increase in diversification rate (0.4 species per million) in *Carex* clades sister to Siderostictae clade (*Spalink et al., 2016a*) and the role of the Himalayas and Hengduan Mountains as engines of global biodiversity (*Acharya et al., 2011*; *Grytnes & Vetaas, 2002*; *Liu et al., 2016*; *Luo et al., 2016*; *Price et al., 2011*; *Xing & Ree, 2017*; *Xie et al., 2014*; *Wang et al., 2012*), our study suggests that the Himalayas have been more nearly an evolutionary dead-end for *Carex* outside of the core Unispicate clade. The abundance of lineages that have dispersed into the Himalayas and failed to diversify (Fig. 1) is remarkable and at odds with our expectations at the outset of the study. It is also somewhat remarkable that dispersal rate out of the Himalayas (0.28 events per million years) is twice as high as diversification rate within the Himalayas (0.16 events per million years), and it is perhaps telling that narrow sense biogeographic coding failed to retrieve any clades endemic to the Himalayas.

It may well be that *Carex* outside of the core Unispicate clade has simply been constrained by competitive exclusion from the Kobresia clade, as has been demonstrated in other Himalayan lineages (*Kennedy et al., 2012*; *Price et al., 2011*; *Price et al., 2014*). This effect may become stronger in the future, as Himalayan plant species are driven primarily uphill by warming climate (*Padma, 2014*). Since ecosystems in the high Himalayas are largely dominated by the Kobresia clade, the opportunity for speciation in other Himalayan *Carex* lineages will be limited, presuming niche conservatism (*Wiens & Graham, 2005*). Thus the importance of the Himalayas to global sedge diversity may well be limited primarily to the contribution of the Unispicate sedges.

## ACKNOWLEDGEMENTS

We thank three anonymous reviewers, whose comments have greatly contributed to improving the manuscript. We thank the staff of The Morton Arboretum and The Field Museum for providing assistance in all aspects of this work. In particular we thank Elisabeth Fitzek and Kevin Feldheim for assistance in the lab; Sadia Malik, Enrique Maguilla, Tamara Villaverde, Richard H. Ree, Nicholas Matzke and Joseph W. Brown for advice on analyses; and Marlene Hahn, Andrea Miller and Lindsey Worcester for assistance with curation of

data and specimens. We are grateful to Dr. Zahid Ullah, and E, MO and MSB herbaria for providing herbarium plant material.

### Funding

This study was funded by Higher Education Commission (HEC), Pakistan to Uzma under International Research Support Initiative Program (1-8/HEC/HRD/2015/3947) and NSF grant (1255901) to Andrew L. Hipp. The funders had no role in study design, data collection and analysis, decision to publish, or preparation of the manuscript.

### Grant Disclosures

The following grant information was disclosed by the authors:
Higher Education Commission (HEC).
International Research Support Initiative Program: 1-8/HEC/HRD/2015/3947.
NSF: 1255901.

### Competing Interests

The authors declare there are no competing interests. Pedro Jiménez-Mejías is an Academic Editor for PeerJ.

### Author Contributions

- Uzma conceived and designed the experiments, performed the experiments, analyzed the data, contributed reagents/materials/analysis tools, prepared figures and/or tables, authored or reviewed drafts of the paper.
- Pedro Jiménez-Mejías contributed reagents/materials/analysis tools, authored or reviewed drafts of the paper.
- Rabia Amir contributed reagents/materials/analysis tools.
- Muhammad Qasim Hayat conceived and designed the experiments, contributed reagents/materials/analysis tools.
- Andrew L. Hipp conceived and designed the experiments, analyzed the data, contributed reagents/materials/analysis tools, authored or reviewed drafts of the paper, approved the final draft.

### DNA Deposition

The following information was supplied regarding the deposition of DNA sequences:
All the new sequences are available at GenBank, the accession numbers are available in Table S6.

### Data Availability

Data files and script/codes are available in Github.
https://github.com/uzma-researcher/Himalayan-Carex-Diversification.
https://github.com/uzma-researcher/Himalayan-Carex-Climatic-niche-Evolution.

## Supplemental Information

Supplemental information for this article can be found online at http://dx.doi.org/10.7717/peerj.6792#supplemental-information.

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
