# Peer review of "Timing and ecological priority shaped the diversification of sedges in the Himalayas"

_PeerJ, doi:10.7717/peerj.6792_

## Round 0.1 · original submission · Major Revisions

Reviewers agreed that the paper is very interesting to understand plant diversification in the Himalayas, a remarkable region in relation to biodiversity and that the sedges are a very important and diversified group. I recommend that the suggestions of the three reviewers be carefully considered, they are here included. Two of them coincided in that analyses of molecular dating should be revised, either better explained or re-analyzed. Another concern is related to the number of terminals in phylogenetic analyses that might be reduced to obtain better support in trees on one hand and in the other to try to represent better taxa from different biogeographic areas to avoid biases toward North America. In addition, please reconsider including figure 1.

Reviewer 1 ·

Basic reporting

1. The text well-structured and is written in reasonably good English with the following minor problems noted:
Grammar, phrasing and word usage
Line 147: species of specimens? 27 specimens representing 21 species is clearer.
Line 149: comprising not bringing
Line 152: plural noun for singular verb: should be “repository… was retrieved”; also line 240-41 should be “data…were retrieved”
Line 168: why does the sentence start with However? This is confusing because it implies a comparison which doesn’t seem to be there.
Line 175: However makes more sense here, but is still unnecessary. Two-step not two-steps amplification.
Line 222: presence or absence, not and absence; a species can’t be both
Lines 280-282: Verb tenses don’t agree within the sentence and anagenetic should not be capitalized. The sentence is also unclear as written.
Line 287: explain rather than explaining
Line 290: use for rather than of
Lines 293-295: The sentence has many grammatical problems and the meaning is not at all clear.
Line 368: another problem with agreement between subject and verb (singular/plural)
Line 371: present tense preferred here
Lines 381-384: noun/verb agreement problems (twice): should be “number…was”
Lines 385-389: This need to be rewritten to improve clarity.
Line 418: The “however” in the middle of the sentence needs to be removed to make the sentence work.
Line 421: “regions” should be indicated as possessive (regions’) or say “ between the lineages of the two regions”. A verb is also missing on this line.
Line 421: Change the comma to a semi-colon. The words after the comma are an independent clause.
Line 485: part of the verb is missing
Line 486 also seems to be missing a verb.
Line 516: presumably you mean “ nor to” rather than “as well as to”
Line 526: verb missing
Line 539: sentence is awkward, but “versus” would be better than “than” for the comparison
Line 547: Use twice as high as instead of twice higher than
Line 553: Remove “all the”
Missing articles (a, an, the):
Lines 150, 237, 270, 278, 280, 282, 293, 422, 555 and elsewhere.
2. The introduction and background are sufficient, interesting, well-referenced and well-organized. The four specific questions that organize the project are clearly presented.
3. The structure conforms to standards in the field. I did not take the time to check detailed compliance with PeerJ rules for formatting sections or for formatting references.
4. The figures are mostly clear and informative, but I noted the following problems:
a. The caption for Fig. 2 appears to be the wrong caption and has numerous grammatical problems. It refers to arrows, circles, blocks and abbreviations that I didn’t see on the figure, and it does not explain the significance of the circles labeled 15 and 30 (presumably dates in Ma?)
b. Fig. 3 caption states speculations as fact, as does the text (see below).
c. Fig. 4 caption has grammatical problems (indeed used instead of a verb; compare should be compared). It would be preferable to have the units for Rate given directly on the figure.
5. Tables are well laid out but there are the following problems:
a. Tabl Table 1 has values that are incorrect for the number of informative sites based on checking the original data in the supplemental files (see details below)
b. Table 2 should give standard deviations on the estimates as well as the mean.
c. Table 3 caption needs punctuation. Last line is unclear. Table 3, line 7: Kobresia not Kobresian. Also, there is no indication of what the * means. Table 3, line 8: I don’t understand what this line means.
6. Raw data were available

Experimental design

1. This original research is within the scope of PeerJ
2. The research questions are clear and relevant, addressing an issue that has not yet been considered for this geographic region.
3. The authors present results from a series of sophisticated analyses designed to rigorously test questions related to evolution and biogeography. The problem is that they do this with a data set that is, in my view, not adequate to achieve their goals.

I see three distinct problems with the data set:

First, only 3 genes are used and two of them are nuclear ribosomal spacers which can sometimes be misleading due to incomplete lineage sorting, divergent paralogues, etc., and almost 50% of the data are missing. In checking the data, it appears that 65 species, including 13/38 of those in the Kobresia clades, which are the main focus of the paper, are represented only by one of the three genes (usually ITS, or sometimes ETS). Furthermore, Genbank numbers are given only for the new data gathered for this study (40 species). The other data come from a Dryad repository from an earlier paper and it is not clear whether these sequences have Genbank numbers.

Second, although the level of variability for the 3 genes appears to be very high in Table 1, these numbers do not appear to be correct. The percentage of informative sites seemed extremely high, so I checked the supplemental data files and found that ETS had 501 informative sites (that is, with a mutation shared by at least two species) rather than the 720 specified in Table 1; ITS had 359 informative sites rather than the 590 specified in Table 1; and matK had only 190 informative sites rather than the 319 specified in Table 1. The numbers given in Table 1 are substantially higher than the total number of variable sites, so it is unclear where they came from. In total, there are only 1050 informative sites for resolving 970 species, so it is not surprising that the support values are weak for most clades even in the combined gene tree (supplemental file S8). The authors state on line 337 that the topology is “supported at most major clades”. While it is true that the five major clades of Carex are well supported, the topology within these clades receives only weak or no support, with the exception of sets of two or three closely related species scattered through the tree, a very few well-sampled mostly North American clades (e.g. the Griseae-Careyanae-Granulares group and the Laxiflorae-Paniceae-Bicolores group), and groups that have been the topic of recent study (e.g. the Racemosae clade and the Spirostachyae-Echinochlaenae clade).

Third, the choice of species appears to be geographically biased toward North America. Based on the color-coding in Fig. 2, a large proportion of the species are from the Nearctic and many of those from the Eastern and Western Palearctic are species that also occur in the Nearctic. Few Neotropical species are represented and most of them also occur elsewhere. Similarly, except for African species in the Siderostictae clade and an early-diverging Core Carex clade (primarily section Indicae), only a few species from the Afrotropic and Indo-Mayalan regions are not also distributed in one or more other regions. Most troubling is that nearly all species coded as Indo-Malayan are also coded as Himalayan, even though there are likely to be species in the subtropical Indo-Malayan region that would not occur in the Himalayan region, but these are not included in the sampling. Carex is diverse in the Eastern Palearctic (hundreds of species, many endemic) but this diversity is not reflected in the sampling. Ideally, inference of origins of the Himalayan species and patterns of dispersal between regions, as well as questions related to speciation and extinction rates in different regions should have complete or at least unbiased representative global sampling of species in proportion to their abundance in each region. My concern is that the bias in the sampling may have a strong influence on the results of the study and undermine confidence in the conclusions.

In short, the initial phylogenetic analysis using all 3 genes shows a topology that is largely unsupported except for the major clades, and the choice of taxa is biased rather than representative of the diversity in all regions. All other analyses depend on the same data and topology. How can the conclusions be valid if the tree topology is not supported, except for the 5 major clades?

Problems with dating: Dates are treated as fixed even though the variability associated with estimates in the papers they cite is high. Dating the crown age was based on two papers that used the same fossil calibration. Curiously, another paper on Cyperaceae with Hipp as a co-author found a younger estimate for the crown age of Carex. That paper (Escudero, M. and Hipp, A. 2013. Shifts in diversification rates and clade ages explain species richness in higher-level sedge taxa (Cyperaceae). American Journal of Botany 100: 2403-2411.) is not cited. Two cited papers by Spalink et al. (2016a,b) using additional fossils within Cyperaceae for dating found an even younger mean estimate for the crown age of Carex. This estimate is dismissed as different on lines 354-357, and not used because “we use a calibration based on a more recent date according to Escudero et al. 2012 and compatible with the fossil record (Jimenez-Mejias et al. 2016)”. This is very confusing because Escudero et al. 2012 use an older fossil for dating than the other papers, not a more recent one, and the older fossil they use is treated by Jimenez-Mejias et al. as doubtfully Carex.

4. Most of the methods are described in enough detail but I noted some serious and not so serious omissions of information that should have been included in Materials and Methods and Results sections. These are detailed here:
• Reference is missing for the R packages on line 248.
• The number of species in each category should be given on lines 258-259, and it should be specified whether the software is sensitive to species sets that are drastically different in size (as they likely are here) and accounts for such differences. From first principles, it seems likely that estimating rates from a large sample would be more accurate than estimating rates from a small sample size and the variances would not be homogeneous.
• On line 330, it states that the “phylogeny obtained from ITS (…) showed phylogenetic incongruence”. It is not clear what it is being compared with and the nature of the incongruence should be specified. Supplemental file S4 shows a very different arrangement of the major clades ((Core Carex + Vignea (except C. gibba)) sister to the Core Unispicate clade with the Schoenoxiphium clade nested within it), but this arrangement has no significant support. Given that, there should be some justification given for combining the ITS data with the other data into a single analysis.
• On line 350-353 it says the data calibrations are “based on the same sources” as those in 3 references. It should be clarified what this means: trees based on the same data (which does not appear to be true), based on the same fossils? Based on the same dates obtained in earlier studies? I checked the references and I could not find a dating analysis in Waterway et al. 2009.
• Lines 373-377 are poorly written to the extent that the meaning is not clear.
• Lines 379-81: It would be useful to show the percentage of them in each major clade here.
• Lines 392-394: variability estimates should be given with the means for the estimated dates. They are certainly large enough to put all of the dates into the Miocene.
• Lines 390-end of paragraph: The mean estimates for the dates are treated as ‘truth’ here despite the known variability of the estimates and the many issues related to dating both in terms of the fossils used and methods followed. This section should be rewritten to acknowledge that the dates are really ranges and that what is being interpreted is the relative order of events. As noted elsewhere, the biased sampling of species in the data set makes all of these reconstructions potentially suspect.
• Lines 454-455: This distribution may be true of the dataset, but there are many species in the Decorae and Indicae clades in subtropical southern China and SE Asia that were not included in the dataset, so this is misleading.

Validity of the findings

Due to the problems listed above with the data set, I cannot say the data are robust and statistically sound. Fairly strong conclusions are drawn but I don't think the data support such strong statements, particularly when the dates are treated as being without error variation.

Lines 509-512: This categorical statement needs to be qualified. Given my various concerns about the data and geographic sampling, I think this is a speculative statement, not a conclusion that is warranted by the data and analyses presented in this manuscript.
Lines 515-520: This statement is also speculative and quite likely incorrect. Sampling omissions from these regions can account for the results, so it is speculative to attribute biological reasons. Carex are quite frequent in SE Asia, often in montane regions, which do not appear to have been sampled for this data set.
Lines 538-540: Comparison with previous estimates of diversification rates in Carex and Cyperaceae should be made somewhere in the discussion (e.g. Escudero and Hipp 2013 and other Escudero papers, Spalink 2016)
Lines 546-549: The higher dispersal rate out of the Himalayas than diversification rate within the Himalayas is at odds with the previous statement that the Himalayas were an evolutionary dead end. This result could also be explained by biased sampling that failed to include an appropriate proportion of Himalayan endemics, which are more difficult to sample because of their endemicity.

Additional comments

This is a very interesting project with some sophisticated analyses and approaches to the issues, but several issues with the data need to be addressed. The tree on which all other analyses are based has so little support except for the five major branches that I don't think the conclusions are valid. I think it would be a stronger paper with fewer species chosen to have an unbiased geographic distribution and more genes to assess the relationships. Data for additional genes is available on Genbank for many of the species in your analysis.

Reviewer 2 ·

Basic reporting

The paper is clearly written in general. But I have some concerns.
Citations need to be checked more carefully. For instance, they cited Xie et al. 2014 for geological history of this region. But none of the authors in this paper are geologist. The format of the citation is not consistent and strange. Some “et al.” are in italic and some are not. They should check the formatting guidance carefully. For instance,” van Hinsbergen et al., 2011; van Hinsbergen et al., 2012” should be van Hinsbergen et al., 2011, 2012.
Some figures are not informative or not nice-looking. In my opinion, fig 1 is not necessary and they could merge fig 3 to fig 2.

Experimental design

The diversification history of Himalayan flora is crucial for our understanding of mountain diversity. The authors investigated the biogeographic and diversification pattern of a diverse group, Carex in this region by using comparative phylogenetic methods. They found that the Himalayan radiation may date back the late Oligocene and the success of Kobresia might due to competitive exclusion. This provides new insights into the diversification of Himalayan flora.

Validity of the findings

there are some technical problems in their analysis which I listed in the comments to authors.

Additional comments

The diversification history of Himalayan flora is crucial for our understanding of mountain diversity. The authors investigated the biogeographic and diversification pattern of a diverse group, Carex in this region by using comparative phylogenetic methods. They found that the Himalayan radiation may date back the late Oligocene and the success of Kobresia might due to competitive exclusion. This provides new insights into the diversification of Himalayan flora.
The paper is clearly written in general. But some issues need to be addressed before publication. My concerns are listed in the following.
1. You used “narrow sense” and “broad sense” for biogeography and GeoSSE analysis, but you did not point out in the GeoSSE method part. I was confused which is which in the Results and Discussion part. For instance, when you say the radiation of Kobresia date back to late Oligocene, do you mean the “broad sense” Himalaya? In my opinion, the diversification history of the broad sense Himalaya might be totally different from the “true” Himalaya. This is because your broad sense Himalaya includes three distinct geological entities, the Himalayas, the Tibetan Plateau and the Hengduan mountains. They have different uplift history. This matters when you interpret the biogeographic history. For instance, you said that the Himalayan Kobresia was dispersed from the Nearctic region, but it is also possible that the “true” Himalayan Kovresia may arise from the Tibetan Plateau as it is older than the Himalayas.
2. For your GeoSSE analysis, you interpret the dA and dB as dispersals in the Himalayas and non-Himalayas. This is not correct. You made it clear that dA means the dispersal rate from the Himalayas to non-Himalayas, not within the Himilayas.
3. Please check your citations more carefully. You cited Xie et al. 2014 for geological history of this region. But none of the authors in this paper are geologist. The format of the citation is not consistent and strange. Some “et al.” are in italic and some are not. When you have references by the same authors from different years, you only list author names once. For instance,” van Hinsbergen et al., 2011; van Hinsbergen et al., 2012” should be van Hinsbergen et al., 2011, 2012.
4. Please replace “Late Miocene” or “Late Oligocene” to “late Miocene” or “late Oligocene” as late Miocene is not a formal geological epoch any more.
5. Some figures are not informative or not nice-looking. In my opinion, fig 1 is not necessary and you could merge fig 3 to fig 2.

Reviewer 3 ·

Basic reporting

no comment

Experimental design

no comment

Validity of the findings

no comment

Additional comments

This is an exciting manuscript on the diversification and biogeographical history of the Himalayan species of Carex. The authors have compiled an excellent dataset to investigate a series of questions related to the origin, adaptation, and rate of diversification Carex lineages in the Himalayas. Their main conclusions are that Himalayan Carex are the result of many instances of migration from outside sources; detectable in situ radiations was largely limited to a single clade; that rates of diversification in the Himalayas was low; arrival of Carex in the Himalayas postdates orogeny and uplift; and lineages that entered the Himalayas were possibly preadapted to alpine-type climates.

The writing is generally clear and the manuscript is easy to follow. My suggestions for improvement are mostly minor; some re-analyses may be required unless the authors can provide sufficient justification for their choices.

Molecular Dating: Given their reliance on a single calibration point for a 900+ taxa tree, I spent a bit of time comparing the various recent Cyperaceae dating analyses. Interestingly, their calibration point is the Carex crown age (42.8 Ma; Mid Eocene) obtained by Escudero et al. 2012, whose sole Carex calibration is a fossil whose veracity has been previously questioned by an author of this study (Carex tsagajanica). If my reading of Jiménez-Mejías et al (2016) is correct, the oldest verifiably Carex fossil is from the late Eocene, with fossils becoming increasingly common into the Miocene and on. Thus, while the date of the Carex crown in this study is consistent with the fossil record, the younger dates obtained by Escudero and Hipp (2013), Spalink et al (2016), and Léveillé-Bourret et al (2018) are not inconsistent with the fossil record.

With the exception of the set 42.8 Ma Carex crown, most other dates on their chronogram are older than previous studies – even those that have used the same calibration, e.g., the Siderostictae crown is 15 Ma in this study, 10 Ma in Spalink 2016, and 5 Ma in Escudero 2012; unispicate clade crown is 30 in this study, 18 in Waterway, 22 in Escudero, and 15 in Spalink; the core Carex clade is 28 in this study, 18 in Waterway, 21 in Escudero, and 15 in Spalink.

This is not to say that the dates in this manuscript are wrong, but as the timing of events is important for many of their conclusions, it would be helpful if the authors could address these concerns and decide if any re-analyses are merited.

Other suggestions:
Line 82, e.g.: sometimes the authors refer to the Tibetan-Plateau, other times to the QTP – are these the same?

Line 104: “This high variation in … vegetation types.” This is the first time vegetational diversity is mentioned – please rephrase or provide background information.

Line 114: “Disproportional” relative to what?

Line 116: “which are sister to smaller clade” – please clarify.

Sentence beginning Line 125: 40 species is not a majority of 189 species – please clarify.

Line 133: it is a bit confusing to refer to “former genus Kobresia.” Since you have already established that Kobresia is no longer a genus – but presumably refers to a meaningful clade – perhaps you could say something like “(hereafter Kobresia) the first time you mention this issue.

Sentence beginning Line 135: Please clarify this sentence – the meaning is unclear.

BioGeoBEARS analysis: please provide a sentence or two describing why the assumptions of DEC (as opposed to DIVE or BAYAREA) were appropriate for your analyses. Also, there do not seem to be any dispersal constraints in this analyses – meaning that migration from South America to the Himalayas is as likely as migration from adjacent China to the Himalayas.

Line 499: I don’t see where the authors tested for competitive exclusion. Please revise.

Line 533: Are there other undersampled sections of the phylogeny that could impact the interpretation of the results?

Line 540: was there any actual statistical significance?

Line 547-8: please provide units.

When referencing preadaptation: it is important to mention that climate is only one aspect of niche.

---

## Round 0.2 · Minor Revisions

Thank you for considering all previous suggestions by the reviewers. There is only one change suggested by reviewer 1. Please consider it for the paper be accepted.

Reviewer 2 ·

Basic reporting

no comment

Experimental design

no comment

Validity of the findings

no comment

Additional comments

The authors did great job in revising the manuscript. The manuscript has been improved substantially. I only have one minor comment.
L472-473: delete ", during which time the Indian Tibetan continent collided with Asian plates, causing the initial uplift of the Tibet plateau". Because the collision between Indian and Asian plantes was much earlier than 25 Ma.

---

## Round 0.3 · accepted · Accept

Thank you for considering all suggestions by the reviewer, your paper looks very good.

#